# Anti-IL-4, Anti-IL-17, and Anti-IFN-Gamma Activity in the Saliva of *Amblyomma sculptum* Ticks

**DOI:** 10.3390/ijms26104734

**Published:** 2025-05-15

**Authors:** Helioswilton Sales-Campos, Chamberttan Souza Desidério, Rafael Obata Trevisan, Rodolfo Pessato Timóteo, Victor Hugo Palhares Flávio-Reis, Yago Marcos Pessoa-Gonçalves, Marcos Vinicius da Silva, Eliane Esteves, Thiago de Jesus Oliveira, Pedro Ismael da Silva Junior, Carlo José Freire Oliveira

**Affiliations:** 1Department of Bioscience and Technology, Institute of Tropical Pathology and Public Health, Federal University of Goiás, Goiania 74605-050, Goiás, Brazil; tonsales@ufg.br; 2Department of Microbiology, Immunology and Parasitology, Institute of Biological and Natural Sciences, Federal University of Triângulo Mineiro, Uberaba 38025-180, Minas Gerais, Brazil; chamberttan_sd@hotmail.com (C.S.D.);; 3Laboratory of Infectious Diseases, Department of Microbiology and Immunology, University of South Alabama, Mobile, AL 36688, USA; 4Laboratory for Applied Toxinology (LETA), Butantan Institute, São Paulo 05585-000, São Paulo, Brazil

**Keywords:** ticks, saliva, cytokine-binding proteins, immunomodulation, *Amblyomma sculptum*

## Abstract

The saliva of hematophagous arthropods, such as ticks and triatomines, contains bioactive ligands capable of modulating immune molecules, including cytokines. Cytokines play a critical role in immune regulation and have therapeutic relevance in inflammatory and immune-mediated diseases. Despite recent advances, identifying cytokine-binding molecules remains a significant challenge. Interferon-gamma (IFN-γ), interleukin-4 (IL-4), and interleukin-17 (IL-17) are key cytokines involved in inflammation, adaptive immunity, and host defense. This study evaluated the ability of salivary components from *Amblyomma sculptum* and compared the results to the triatomine *Rhodnius neglectus* (used as control) to bind to IL-2, IL-4, IL-6, IL-10, IL-17, IFN-γ, and TNF-α using ELISA assays with human cytokines. Saliva samples were tested at dilutions of 1:25, 1:50, and 1:100. Saliva from *A. sculptum*, which demonstrated significant anti-cytokine activity, was fractionated via HPLC to identify the active components. The results confirmed the inhibitory capacity of *A. sculptum* saliva on IFN-γ, IL-4, and IL-17, with inhibition rates ranging from 30% to 70%, depending on the cytokine and dilution. No inhibitory activity was observed against IL-2, IL-6, IL-10, or TNF-α. These findings underscore the immunomodulatory role of *A. sculptum* saliva during tick feeding and suggest its potential for the development of novel immunobiologics to treat inflammatory and immune-mediated diseases.

## 1. Introduction

Historically, natural compounds derived from plants, herbs, animals (such as mosquitoes and ticks), and microorganisms have been reliable sources of bioactive molecules for drug discovery. The emergence of new technologies, along with the development of high-throughput screening methods, has greatly accelerated the discovery of novel biological activities and the identification of active components from these natural sources [1,2,3]. Among these, bioactive molecules derived from animals, particularly hematophagous arthropods and parasites, have garnered increasing interest [4]. Ticks represent a rich reservoir of pharmacologically active molecules, owing to their ability to overcome the physical, inflammatory, hemostatic, and immunological barriers of their hosts [5]. As obligate hematophagous parasites, ticks must evade the defense systems of their vertebrate hosts to successfully feed and survive—an effort in which saliva plays a central role. Tick saliva contains a diverse array of bioactive molecules that interfere with blood coagulation, platelet aggregation, vasoconstriction, inflammation, and both innate and adaptive immune responses of the host [6]. Several studies have identified molecules capable of modulating the host immune system, including evasins, which have been characterized as ligands for chemokines, histamines, antibodies, and other immune mediators [7].

Cytokines, a broad family of signaling molecules that includes chemokines are pivotal regulators of host immunity and play critical roles in protective responses against ticks. Given their importance, it is plausible that ticks produce and secrete molecules capable of binding to cytokines and chemokines, thereby modulating host immune responses. Although several studies have investigated the effects of tick feeding and saliva on cytokine expression and activity, further research is needed to identify the specific salivary components involved in this modulation [8,9], the specific mechanisms and roles of these interactions remain poorly understood. Some cytokine-binding effects of tick saliva molecules have been demonstrated, targeting molecules such as IL-8, MCP-1, MIP-1α, RANTES, eotaxin, and IL-2 [10]. Furthermore, recent studies have identified molecules with anticytokine activity, including those targeting IL-4, TNF-α, and IFN-γ. For example, anti-IL-4 activity has been observed in salivary gland extracts from *Ixodes ricinus* and *Rhipicephalus appendiculatus* nymphs, although the specific molecules responsible and their mechanisms of action remain unknown. Similarly, anti-TNF-α activity has been detected in the saliva of ticks from the genera *Ixodes* and *Haemaphysalis*, but not in species of *Rhipicephalus*, *Dermacentor*, or *Amblyomma* [11].

These findings highlight the species-specific nature of cytokine-binding activities, which are likely shaped by evolutionary pressures associated with host–parasite interactions. However, research has largely focused on a limited number of tick species from North America and Europe. Globally, over 950 tick species have been identified, with approximately 200 occurring in Neotropical regions. Brazil alone is home to around 70 species [12,13], highlighting the potential to discover novel cytokine-binding molecules in these regions. *Amblyomma sculptum* is one of the most important tick species in Brazil, parasitizing a wide range of hosts, including horses, capybaras, cattle, cats, and humans, and acting as the primary vector of Brazilian spotted fever [14]. Although several studies have demonstrated the anti-inflammatory and immunomodulatory activities of its saliva, molecules with cytokine-binding properties have yet to be identified and characterized in this species [15,16,17]. Triatomine saliva also exhibits immunomodulatory properties. Among these insects, *Rhodnius neglectus* is a hematophagous species of medical relevance, recognized for its role in transmitting *Trypanosoma cruzi*. Despite its epidemiological importance, little is known about the immunomodulatory potential of its salivary components. In addition, based on the observation that this triatomine parasitizes a smaller number of hosts compared to *A. sculptum*, and is therefore exposed to less environmental pressure than the tick species, the triatomine *R. neglectus* was used as a control in this study.

Thus, the present study aimed to evaluate the ability of salivary molecules from *A. sculptum* and the triatomine *R. neglectus* to bind cytokines involved in innate and adaptive immunity. These cytokines play critical roles in tick–host interactions and identifying molecules that can bind to them may offer new insights into the immunomodulatory strategies employed by these arthropods, as well as the potential therapeutic use of such binding molecules in treating inflammatory and immune-mediated diseases.

## 2. Results

### 2.1. Amblyomma sculptum Saliva Binds to Human IL-4, IL-17, and IFN-γ Cytokines

The first step was to evaluate the ability of *A. sculptum* saliva to bind the human cytokines IL-4, IL-17, and IFN-γ. IL-4 was effectively neutralized by *A. sculptum* saliva in a dilution-dependent manner (Figure 1A). A decrease in anti-cytokine binding activity was observed with increasing saliva dilution (Figure 1A). Specifically, the 1:25 dilution inhibited IL-4 binding by 36.97%, while the 1:50 and 1:100 dilutions resulted in inhibition rates of 25.19% and 17.54%, respectively.

Similarly, *A. sculptum* saliva neutralized IL-17 (Figure 1B) and IFN-γ (Figure 1C) at all tested dilutions. For IL-17, inhibition rates were 40% at 1:25 dilution, 19.52% at 1:50, and 6.87% at 1:100. In the case of IFN-γ, the saliva showed a strong neutralizing effect, with inhibition rates of 75.5% at 1:25 dilution, 41.2% at 1:50, and 37.99% at 1:100 (Figure 1C).

### 2.2. Amblyomma sculptum Saliva Does Not Show Effective Binding Activity Against Cytokines IL-2, IL-6, IL-10, and TNF-α

To assess whether the salivary effect was specific and consistent across a broader range of cytokines, we tested its activity on an additional group of cytokines. Anti-TNF-α binding activity was detected only at the 1:25 dilution, with a modest reduction of 3.17% (Figure 2D). For IL-6, binding activity was observed at both 1:25 and 1:50 dilutions, with inhibition rates of 4.96% and 4.09%, respectively (Figure 2B). In the case of IL-10, salivary activity was 5.67% at 1:25 dilution and 4.28% at 1:50 dilution (Figure 2C). Similarly, for IL-2, the saliva showed 7.14% inhibition at 1:25 dilution and 0.86% at 1:50 dilution. No detectable activity was observed for any of the tested cytokines at the 1:100 dilution (Figure 2A).

### 2.3. Anti-IL-4, Anti-IL-17, and Anti-IFN-γ Binding Activity in A. sculptum Tick Saliva Fractions

Following the fractionation of *A. sculptum* saliva, the anti-cytokine activity of the resulting fractions was assessed exclusively for cytokines that had shown significant responses in earlier assays. For IL-4, the greatest number of active fractions and the highest levels of inhibition were observed. Eleven fractions (3, 5, 8, 9, 10, 11, 12, 31, 41, 49, and 51) demonstrated inhibition levels above 20% (Figure 3A). Among these, fraction 9 exhibited 40.41% inhibition, fraction 10 reached 64.38%, and fraction 11 showed 51.36%.

In contrast, IL-17 displayed a less pronounced response, with only three fractions, fraction 9 (12.47%), fraction 20 (11.18%), and fraction 52 (11.18%), showing inhibition levels above 10% (Figure 3B).

Consistent with results obtained using whole saliva, the inhibitory effects of tick saliva fractions on IFN-γ were more prominent than those observed for IL-17. Seven fractions (23, 31, 37, 38, 48, 49, and 51) showed inhibition percentages above 10%, with three of them standing out for their stronger effects: fraction 23 (27.45%), fraction 38 (20.78%), and fraction 51 (22.35%) (Figure 3C).

### 2.4. The Anti-Cytokine Binding Activity Was Not Evidenced in Triatomine Saliva

To assess whether the observed anti-cytokine binding activity was specific to ticks, we tested the saliva of the triatomine *Rhodnius neglectus*, a common hematophagous arthropod species in Brazil. No significant effects were observed for IL-6, IL-17, and TNF-α (Figure 4C,E,G). Although minor variations were noted for IL-2, IL-4 and IFN-γ (Figure 4A,B,F), no measurable anti-cytokine activity was detected for these cytokines. The only cytokine for which *R. neglectus* saliva demonstrated a significant and measurable inhibitory effect was IL-10, with inhibition rates of 3.75% and 3.83% at the 1:25 and 1:50 dilutions, respectively (Figure 4D). These findings reinforce our previous observation that, due to greater environmental pressure and a broader range of parasitized hosts, and ticks, specifically *A. sculptum* serves as a richer reservoir of anti-cytokine binding molecules compared to triatomines such as *R. neglectus*.

## 3. Discussion

Tropical countries harbor a vast diversity of hematophagous arthropods, including mosquitoes, sandflies, triatomines, and ticks, among others. These organisms have developed remarkable adaptability to their hosts throughout thousands of years of co-evolution, enabling them to efficiently evade or manipulate host defense mechanisms. Among them, mosquitoes (e.g., *Aedes aegypti*), sandflies (*Phlebotomus* spp.), and triatomines (*Triatoma infestans*) have been extensively studied for their role in transmitting pathogens such as arboviruses, *Leishmania*, and *Trypanosoma cruzi*. However, ticks stand out as a particularly rich source of pharmacologically active molecules, owing to their prolonged feeding behavior, which necessitates overcoming physical, hemostatic, inflammatory, and immune barriers. Tick saliva has demonstrated notable bioactive properties, including the modulation of immune cells and soluble molecules essential to host defense [16,18].

It is important to note that the *A. sculptum* ticks used in this study were collected from horses, a common and natural host of this species. However, it is well established that the vertebrate host can influence the expression of salivary proteins in ticks, including those with immunomodulatory functions. Host-dependent modulation of the salivary gland transcriptome and proteome has been reported in several tick species, potentially impacting the bioactivity of tick saliva. Moreover, variations in salivary composition are not limited to host species; they also occur across different feeding stages and in response to pathogen presence. For example, the saliva of *Rhipicephalus microplus* differs markedly between partially and fully engorged females, reflecting adaptations to distinct phases of blood feeding [19,20]. Therefore, although our results demonstrate cytokine-binding activity in saliva obtained from horse-fed ticks, we acknowledge that variations may arise when ticks feed on different vertebrate hosts. Future comparative studies involving ticks fed on various hosts, such as rodents, livestock, or humans, could offer valuable insights into the consistency and variability of these salivary effects on host immune responses. Ticks are among the most extensively studied hematophagous arthropods due to their remarkable adaptability to a wide range of hosts throughout their life cycle. This host diversity, combined with their prolonged feeding habits, makes ticks ideal subjects for studying bioactive molecules with immunomodulatory activity. Their saliva exhibits fascinating bioactive properties, including the modulation of both immune cells and soluble molecules critical to host defense. The results presented here, demonstrating the anti-binding activity of *A. sculptum* saliva and its fractions against the cytokines IL-4, IL-17, and IFN-γ, contribute to a growing body of evidence that highlights the role of tick saliva in modulating the host immune response [5,15,21].

Regarding IL-4, we observed the highest levels of modulation of anti-binding activity against IL-4, both by raw saliva at different dilutions and by isolated saliva fractions. This finding is closely linked to the critical role of IL-4 in the immune response during tick infestation, particularly in the balance between resistance and susceptibility to ticks. IL-4 is central to the activation of mast cells and basophils, which are key mediators of inflammatory responses such as urticaria and pruritus—responses commonly seen in tick-resistant hosts. These inflammatory reactions hinder tick attachment and feeding, as demonstrated in studies where tick-resistant hosts exhibited elevated IL-4 production. In experiments with mice infested with nymphal ticks, it was observed that, upon in vitro stimulation with increasing doses of tick antigens, T cells from the draining lymph nodes showed an initial increase in proliferation and IL-4 production, which peaked and then decreased [6,22,23]. There is evidence that the modulation of IL-4 by tick saliva is not exclusive to *A. sculptum*, as a protein such as sphingomyelinase from *Ixodes scapularis* was the first identified tick molecule capable of programming host CD4(+) T cells to express IL-4 [24]. Thus, tick salivary molecules’ ability to modulate IL-4 activity, through anti-IL-4 binding activity, as demonstrated in our work, or through inhibition of activity, as demonstrated in other works, represents a sophisticated strategy to suppress inflammatory responses that would otherwise hinder tick attachment, facilitate prolonged blood feeding, and allow pathogen transmission.

Regarding IL-17, we observed reduced cytokine levels through inhibition of the binding capacity of diluted saliva and some isolated fractions, although with less significant inhibition of binding activity. However, there is a noticeable research gap regarding the anti-IL-17 activity of *A. sculptum* saliva and other tick species, especially in relation to their role in infestation. Most studies focus on the cellular changes induced by salivary proteins, leaving the immunomodulatory effects on cytokines underexplored. Studies have shown a decrease in IL-17 expression following the addition of tick saliva two hours after incubating dermal cells exposed to *Borrelia burgdorferi*, a finding that aligns with our results on anti-cytokine activity, even if not through the direct binding of salivary molecules to the cytokine [17]. This anti-IL-17 potential warrants further investigation, given that IL-17 plays a central role in bridging innate and adaptive immunity, with significant implications for health and disease. While IL-17 is essential for defending against extracellular pathogens such as bacteria and fungi, its dysregulation has been implicated in a range of inflammatory and autoimmune diseases, including psoriasis, rheumatoid arthritis, and inflammatory bowel disease. Tick saliva could therefore serve as a valuable tool for modulating IL-17 in the contexts of these diseases [25,26].

IFN-γ plays a central role in driving Th1-mediated immune responses, which are essential for controlling intracellular pathogens. This response can be influenced by tick saliva, as studies have shown that both tick and sandfly saliva can alter the pathogen clearance capacity of macrophages. Specifically, it has been demonstrated that the gland lysate of *Phlebotomus papatasi* inhibits the killing of *Leishmania major* parasites by infected murine macrophages [27]. In ticks, the saliva of *Rhipicephalus sanguineus* has been shown to reduce the killing of intracellular *Trypanosoma cruzi* forms by IFN-γ-activated macrophages. Moreover, the inhibition of IFN-γ-induced trypanocidal activity by macrophages, mediated by saliva, was accompanied by a 69% reduction in nitric oxide (NO) production. Additionally, tick saliva increased IL-10 production two-fold and reduced IFN-γ production by 84.6% in splenocytes cultured with *T. cruzi*, suggesting that the decreased NO production by macrophages may result from a cytokine imbalance induced by saliva [28]. In most studies, the results primarily focus on the effects of saliva or salivary gland extract (SGE) on immune cells. For example, it has demonstrated the suppression of IFN-γ production by immune cells, such as macrophages and lymphocytes, mediated by the salivary gland extract (SGE) of *Dermacentor andersoni* [29]. Additionally, molecules already identified, such as cystatin sialostatin L2, have demonstrated the ability to inhibit IFN-γ production in murine dendritic cells. Future studies should focus on evaluating the modulatory effects of saliva, or its isolated compounds, on the activity of IFN-γ and other cytokines, whether through direct or indirect mechanisms, since these molecules play key roles in the host’s response to various aggressors, including ecto-parasites [30].

One important consideration regarding these changes is the species-specific adaptability of ticks, which plays a critical role in their ability to modulate the host’s immune system. *A. sculptum*, for example, is a tick species with a broad range of hosts, including horses and capybaras. This host variability, combined with a high level of adaptability, may enhance its ability to produce diverse bioactive molecules capable of overcoming the immune defenses of various hosts. In comparison to other tick species with identified neutralizing molecules, such as *Ixodes ricinus* or *Rhipicephalus microplus*, the host diversity of *A. sculptum* may suggest a similarly broad or even unique repertoire of immunomodulatory molecules [14,31,32]. Future studies comparing the molecular profiles of different tick species will be crucial to elucidate how host diversity shapes the evolution of these molecules.

Despite our promising results, we acknowledge certain limitations in our study. Notably, these include the use of ELISA to investigate cytokine/chemokine-binding molecules within the fractions, as well as the lack of quantification and characterization of the molecules present in each fraction. In the absence of additional controls specifically designed to test for Fc binding, our current interpretation of the results assumes that the observed signals arise from specific interactions between the saliva components and the cytokines or chemokines of interest. However, this assumption cannot be confirmed without the implementation of proper controls.

Our findings reveal a significant reduction in the levels of cytokines IL-4, IL-17, and IFN-γ by *A. sculptum* saliva or its fractions, probably due to the anti-binding capacity that may affect its biological activity. These results corroborate previous studies that have characterized similar binding interactions with various cytokines. Earlier research documented the anti-cytokine activity of TNF-α, highlighting its inhibition at different concentrations and observing this activity across distinct species [11]. TGF-β has been shown to be susceptible to binding by molecules present in the saliva of Ixodidae ticks. A robust study from this group also demonstrated the binding capacity of salivary gland extracts (SGE) from three species of Ixodid ticks, *Dermacentor reticulatus*, *Amblyomma variegatum*, and *Ixodes Ricinus*, highlighting their ability to bind several cytokines, including IL-8, MCP-1, MIP-1α, RANTES, eotaxin, IL-2, and IL-4 [10,33]. These findings emphasize the importance of identifying and characterizing the specific salivary molecules responsible for these effects. Understanding the molecular mechanisms underlying these interactions will not only deepen our knowledge of the tick–host relationship but also open new possibilities for developing immunobiologicals aimed at treating immune-mediated diseases.

## 4. Materials and Methods

### 4.1. Tick and Triatomine Saliva

The saliva from wild *A. sculptum* was obtained using the technique described by Castagnolli and collaborators (2008) [34]. Engorged females (teleogines) were collected during natural infestations on horses in Uberaba, MG, Brazil. After collection, the ticks were immediately transported alive to the laboratory, with the transportation time not exceeding one hour. The saliva extraction process was carried out shortly thereafter. For saliva extraction, the ticks were secured on adhesive tape and kept in a humid chamber at a temperature above 25 °C. Prior to subcutaneous inoculation into the hemocoel with a 0.2% dopamine solution diluted in PBS, the ticks were cleaned using 0.01 M phosphate-buffered saline (PBS) (Monobasic Sodium Phosphate, Dibasic Sodium Phosphate, Sodium Chloride, and Potassium Chloride; pH 7.2). Saliva secretion from the hypostome was collected using a micropipette with sterile tips, and the saliva was stored at −80 °C until use. Upon thawing, the saliva was diluted into three different concentrations to assess the effects of the molecules at various concentrations. Saliva collection was conducted for 1.5 h at a temperature of 37 °C.

Saliva from adult *Rhodnius neglectus* triatomines was collected using the method described by Mesquita et al. [35], with modifications. Adult insects of the genus *Rhodnius*, of both sexes and subjected to fasting periods of 7 to 21 days, were immobilized on ice, then cleaned with distilled water and 70% ethanol to ensure sterility. To collect the salivary glands, the heads of *Rhodnius* specimens were carefully detached using sterilized forceps, which facilitated the exposure and subsequent extraction of the glands. This procedure was conducted under a stereomicroscope to ensure precision and avoid contamination. The extracted glands were kept on ice throughout the process. Every three pairs of glands (from three insects) were pooled in 10 µL of sterile saline. To extract saliva, the collected salivary glands were pierced with sterile needles, allowing the saliva to leak out. The samples were then centrifuged at 11,000× g for 5 min to remove debris, and the supernatant containing the saliva was carefully collected. The resulting saliva samples were stored at −80 °C until further use.

For *A. sculptum*, the total amount of saliva used in this study was pooled from 42 ticks, resulting in a total protein concentration of 14.5 mg/mL. For *R. neglectus*, 62 pairs of salivary glands from 31 specimens were used, yielding a final concentration of 31 mg/mL. Both total protein concentrations were estimated using the Bradford assay, as previously described [36]. In addition, protein concentration was further confirmed by spectrophotometry using NanoDrop (Thermo Fisher Scientific, Waltham, MA, USA), and the results were comparable between these two techniques.

### 4.2. Binding Effect of Tick and Triatomines Saliva Molecules on Cytokines

The ability of molecules in tick saliva to bind to cytokines was assessed using ELISA assays, a standard method commonly performed in immunology laboratories. This technique is widely used and was previously described under similar conditions [10,11,33]. Human Duoset ELISA kits (R&D Systems, Minneapolis, MN, USA) were employed to examine interference in the production of the following cytokines: IL-2, IL-4, IL-5, IL-17, IFN-γ, and TNF-α. Assays were carried out according to the manufacturer’s protocol. However, prior to adding the recombinant cytokine, it was pre-incubated for 30 min with 50 µL of different dilutions (1/25, 1/50, and 1/100) of *A. sculptum* and *R. neglectus* saliva, or the isolated fractions of *A. sculptum* saliva.

Briefly, high-affinity ELISA plates (Corning-Costar, Corning, NY, USA) were coated with a specific monoclonal antibody (50 μL) for the cytokines of interest, diluted in the specified buffer, and incubated for 8–12 h at 4°C. The plates were then washed three times with PBS/Tween 0.05% (PBS-T) using an automatic plate washer (ImmunoWash, BioRad Laboratories^®^, Hercules, CA, USA) and blocked at room temperature for 1 h with 200 μL of “assay diluent” (R&D Systems^®^, San Diego, CA, USA). After washing, 50 μL of saliva samples (pre-incubated with recombinant cytokine proteins, in triplicate) or standard cytokine curve samples (in duplicate) were added and incubated for 2–3 h at room temperature. Following this, the plates were washed again, and 50 μL of biotinylated secondary antibody was added for 1 h of incubation at room temperature. The plates were washed once more and incubated with avidin-conjugated peroxidase (1.25 ng/mL) (Genzyme Diagnostics^®^, Cambridge, UK) diluted in “assay diluent” for 20 min at 37 °C. After three additional washes with PBS, 50 μL of tetramethylbenzidine (TMB—Sigma^®^, St. Louis, MO, USA) reagent was added to develop the reaction. The reaction was stopped with 2 M sulfuric acid, and absorbance was measured at 450 nm using an ELISA reader (EMAX, Molecular Devices Corporation, Sunnyvale, CA, USA).

Cytokine concentrations were estimated by comparing the results with a standard curve made using the respective murine or human cytokines. The final concentration was estimated using absorbance data analyzed by linear regression using a curve analysis program (GraphPad Prism^®^ 8.0, San Diego, CA, USA).

### 4.3. HPLC Procedures

*A. sculptum* saliva (1.5 mL) was centrifuged through a YM-5 microcon device (Millipore, Bedford, MA, USA). The filtered <5 kDa fraction was submitted to a reversed-phase HPLC on a C18 column (4.3 × 150 mm; ThermoSeparation Products, Riviera Beach, FL, USA) perfused at 0.5 mL/min using a CM-4100 pump (Thermo Separation Products, Riviera Beach, FL, USA). The eluent was monitored at 220–500 nm using a diode array detector (model SPD M10AV, Shimadzu, Columbia, MD, USA). A gradient of 5 to 80% acetonitrile containing 0.1% trifluoroacetic acid, during 80 min was imposed after injection of the sample. Aliquots of these fractions were dried in 96 well plates and tested for cytokine inhibitory activity at dilutions compatible for those employed for crude saliva.

### 4.4. Activity of Saliva Fractions on Cytokines

The *A. sculptum*’s saliva fractions obtained by HPLC were tested exclusively for the cytokines that exhibited inhibitory activity with the diluted total saliva. The assay was performed as described in Section 4.2, with the exception that, in this case, the recombinant cytokines were incubated together with the saliva fractions.

### 4.5. Data Analysis

In all experiments, samples were tested in triplicates. The significance obtained between experimental groups was evaluated by analysis of variance (one-way ANOVA) and Tukey post hoc test using GraphPad Prism^®^ 8.0, (San Diego, CA, USA). The results were considered significant when *p* < 0.05. Data are shown as mean ± S.E.

## 5. Conclusions

*A. sculptum* saliva, together with its fractions, exhibits the ability to exert anti-cytokine binding activity, leading to a reduction in cytokine detection, which was not observed for the triatomine species used in the present study as a control. These findings suggest the presence of a new class of molecules, in addition to evasins (chemokine-binding molecules), with potential immunomodulatory properties. Such molecules may have promising applications in the treatment of several pathologies, including autoimmune and inflammatory diseases that involve the regulation of modulated cytokines.

## Figures and Tables

**Figure 1 ijms-26-04734-f001:**
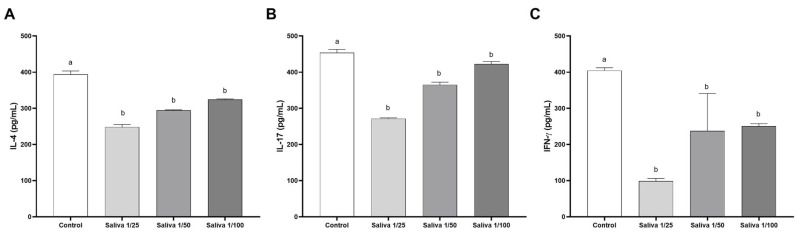
Inhibition of (**A**) IL-4, (**B**) IL-17, and (**C**) IFN-γ binding by *Amblyomma sculptum* saliva. The cytokine binding levels were compared to control levels (white bar) using triplicate values and are represented in pg/mL at saliva dilutions of 1/25 (light gray bar), 1/50 (gray bar), and 1/100 (dark gray bar). Statistical analysis was performed using one-way ANOVA followed by Tukey’s post hoc test. Bars labeled with different letters (e.g., “a” and “b”) indicate statistically significant differences between groups with *p* < 0.05.

**Figure 2 ijms-26-04734-f002:**
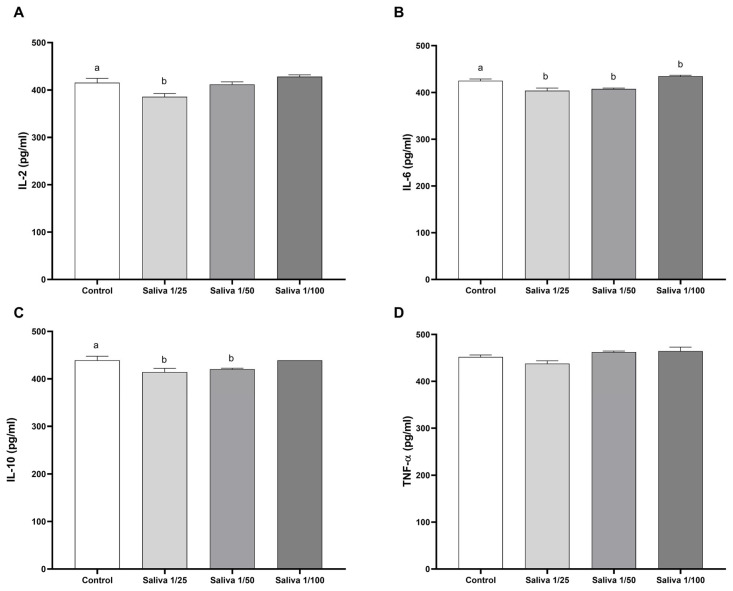
Inhibition of (**A**) IL-2, (**B**) IL-6, (**C**) IL-10, and (**D**) TNF-α binding activity of *Amblyomma sculptum* saliva. The cytokine binding levels were compared to control levels (white bar) using triplicate values and are represented in pg/mL at saliva concentrations of 1/25 (light gray bar), 1/50 (gray bar), and 1/100 (dark gray bar). Statistical analysis was performed using one-way ANOVA followed by Tukey’s post hoc test. Bars labeled with different letters (e.g., “a” and “b”) indicate statistically significant differences between groups with *p* < 0.05.

**Figure 3 ijms-26-04734-f003:**
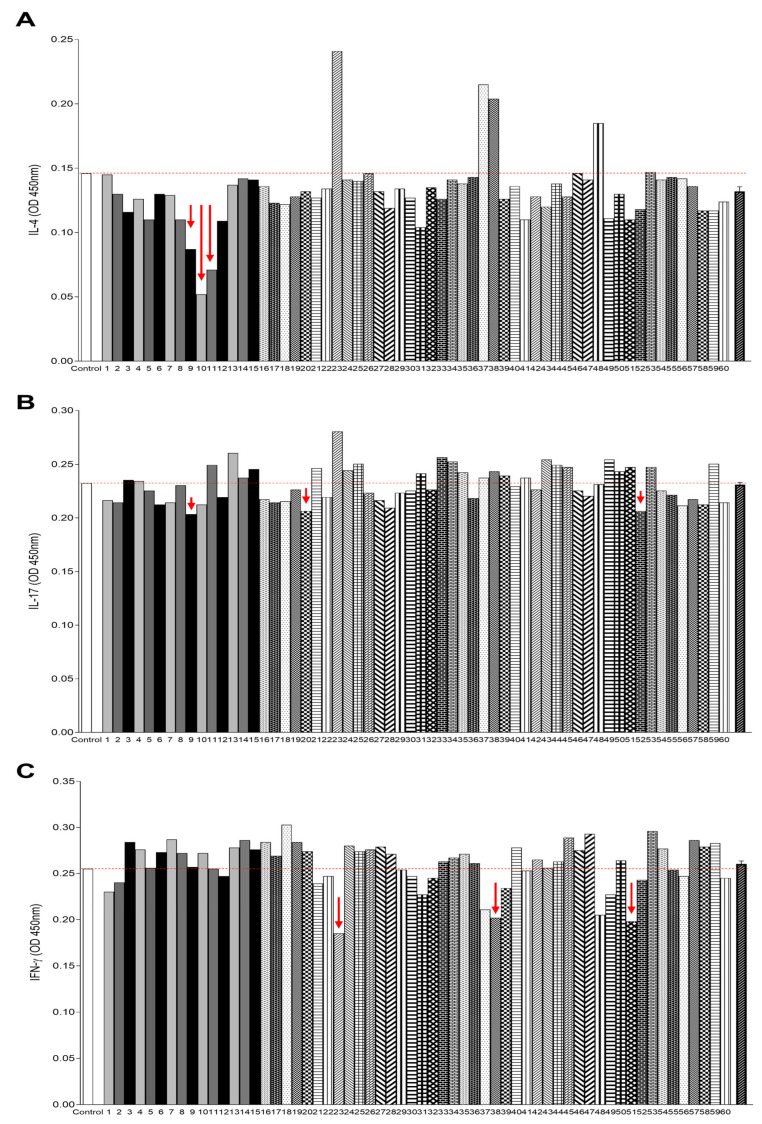
Anti-binding IL-4 (**A**), anti-binding IL17 (**B**), and anti-binding IFN-γ (**C**) activity of *Amblyomma sculptum* saliva fractions. Cytokine binding levels of each fraction are compared. Red arrows highlight fractions with higher anti-cytokine activity. The dashed line serves to better visualize the reduction in cytokine level and is aligned with the control result.

**Figure 4 ijms-26-04734-f004:**
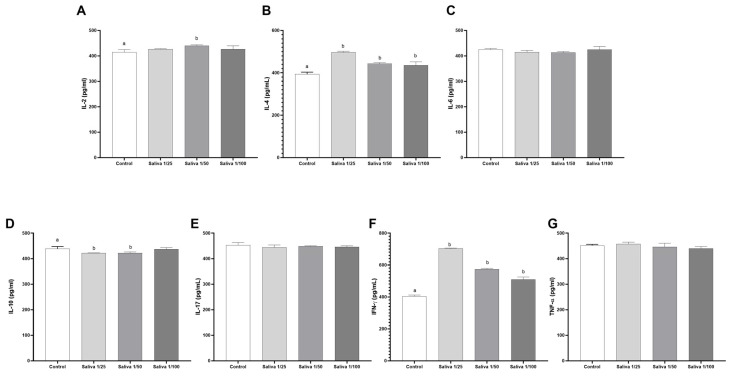
Anti-cytokine binding activity of *Rhodnius neglectus* saliva against: (**A**) IL-2, (**B**) IL-4, (**C**) IL-6, (**D**) IL-10, (**E**) IL-17, (**F**) IFN-γ, and (**G**) TNF-α. The cytokine binding levels were compared to control levels (white bar) using triplicate values and are represented in pg/mL at saliva concentrations of 1/25 (light gray bar), 1/50 (gray bar), and 1/100 (dark gray bar). Statistical analysis was performed using one-way ANOVA followed by Tukey’s post hoc test. Bars labeled with different letters (e.g., “a” and “b”) indicate statistically significant differences between groups with *p* < 0.05.

## Data Availability

Data is contained within the article.

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
