# Peer review of "Anti-IL-4, Anti-IL-17, and Anti-IFN-Gamma Activity in the Saliva of Amblyomma sculptum Ticks"

_ijms, 2025, doi:10.3390/ijms26104734_

Round 1
Reviewer 1 Report
Comments and Suggestions for Authors
The authors report on a study titled “Anti-IL-4, anti-IL-17 and anti-IFN-gamma activity in the saliva 2 of Amblyomma sculptum ticks”.
General comments
I have carefully reviewed the manuscript. It is well written and have made a few comments which are of editorial nature.
Methodology
The Methodology is concise and reproducible; however, the authors need to justify why they included the triatomine bug in their studies both in the Introduction section as well as in the methodology section. There is also need for more information on the number of ticks used in the study and the amount of saliva generated by the ticks for the study.
Results
The data presentation needs improvement and have highlighted directly in the manuscript where improvement can be made.
Discussion
Taking into consideration that the source of ticks was horses. The authors need to discuss if they were going to get same results if source of ticks was from another vertebrate species which is not a horse.
I have attached an annotated copy of the manuscript with comments/suggestion/edits for the authors’ consideration.

The quality of English is acceptable.
Reviewer 2 Report
Comments and Suggestions for Authors
The paper from Sales-Campos and collaborators is potentially very interesting, since it looks for biomolecules in two blood-feeding insects active against human chemokines and interleukines. The aim and the importance of the research are very well described, however there are many points which make the manuscript not suitable for publication as it is.
First of all, the authors claim that the whole saliva or its fractionated products have anti-IL activity, however the ELISA test is not an activity test, but a binding test. Therefore what the authors have presented in their figures is a binding activity of the saliva products with respect to the IL of the kit. The fact that a binding occurs does not mean that it will inactivate a signalling function. None of the results show a functional assay.
In the method section, the Authors claim that the saliva and the fractions have been incubated with either mouse or human interleukins or interferons, but it is not specified which one of the proteins is from mice and which one is from human.
In the figure legends there are some "a" and "b" on top of some histogram, but there is no explanation on their significance.
In the saliva samples, there is no quantification of the material, a reader does not know whether the triplicate have the same amount or not. Moreover since the method of saliva extraction from the 2 different types of insect is diverse, without a quantification a reader cannot know whether the tick is really more effective or if this is due to a higher content of the active biomolecules.
It is also quite strange that the binding values of the triatomine's saliva are higher than the control and this is not at all discussed by the Authors.
In the paragraph text explaining data in Figures 1 and 2 a dilution-dependent effect is clear from the histograms, however the Authors claim that the binding does not depend on the quantity ("regardless dilution").
The binding of IFN-gamma by the fractions is less pronounced than the one by the whole saliva, while for the IL4 a different effect is found. For the fractionation two different protocols have been reported, but it is not written which method has been used with which saliva sample.
Then there are some minor points: in lines 30, 77, 165, 166, 167, 197, 209, 222, 224 and 228 the Lynneus species names are not in italics; throughout the text there are typos such as "ex-pression" in line 61; line 46 gained instead of garnered.
Round 2
Reviewer 2 Report
Comments and Suggestions for Authors
The revised manuscript by Sales-Campos and collaborators has somehow improved with respect to the first submission. However there still remains some crucial points which need to be addressed before publication. In particular, in the methods section there are two different paragraphs dedicated to HPLC fractioning (4.2, 4.4), however it is not stated which one has been used for which type of saliva, nor the quantity of injected material; moreover the Authors do not show any elution profile.
In the ELISA test performed with those fractions (paragraph 4.3 and figure3) it is not stated how much of each fraction has been incubated with the pertaining interleukins and chemokines. The problem is the following: if the binding of one fraction is tightest because there is more quantity of the active, the result might be a false positive. This risk is not negligible especially because the fractionated saliva has not been tested by mass spectrometry or NMR for identification.
Even if this is a preliminary study, quantity matters. The Authors have quantified the protein content by Bradford method, however they inject for fractionation only small metabolites, since they say to take the filtrate of 5 kDa MWCO. This flow-through might contain very few peptidic content, if any. Hence a different quantification methods should have been provided in order for a good comparison to be made. A problem of quantification could also explain why most of the fractions in figure 3 panels B and C give an ELISA signal higher than the baseline control, which should only contain the pertained IL / IFN. The same holds for Figure 3A in which 4 fractions have double the signal than the control and this is not even analysed by the Authors.
This same behaviour is presented with the whole saliva from the triatomine R. neglectus in Figure 4B and 4F. Is this saliva content binding the Fc part of the conjugated Ab-peroxidase used in the ELISA test? A control needs to be performed and added to understand this behaviour, which is concentration-dependent.
In the discussion as well as in the conclusion, the Authors keep on confusing binding with activity. What they measure is specific binding, since none of the IL/TNF/INF has direct activity, such could be an enzyme. The components of the saliva are able to specifically bind some IL, so that the free quantity in the ELISA test is lower than the control. This binding might prevent the IL/IFN/TNF from reaching their receptor and then exert their intracellular activity, but this is not tested nor analysed by the Authors with this binding assay.
All the minor concerns have been addressed in this revised version.
Round 3
Reviewer 2 Report
Comments and Suggestions for Authors
The Authors have answered to the questions as honestly as they could. Indeed more strong controls would have been important to be added to the manuscript, but it can be accepted as a preliminary report.